# An Improved VGG16 Model for Pneumonia Image Classification

Zhi-Peng Jiang [1,2], Yi-Yang Liu [1,3], Zhen-En Shao [1] and Ko-Wei Huang [1,*]

[1]  Department of Electrical Engineering, National Kaohsiung University of Science and Technology, Kaohsiung City 807, Taiwan; i109154106@nkust.edu.tw or cpchiang@vghks.gov.tw (Z.-P.J.); I108154101@nkust.edu.tw (Y.-Y.L.); kotori22339@gmail.com (Z.-E.S.)
[2]  Department of Information Management, Kaohsiung Veterans General Hospital, Kaohsiung City 813, Taiwan
[3]  Department of Urology, Kaohsiung Chang Gung Memorial Hospital and Chang Gung University College of Medicine, Kaohsiung City 833, Taiwan
*  Correspondence: elone.huang@nkust.edu.tw

**Abstract:** Image recognition has been applied to many fields, but it is relatively rarely applied to medical images. Recent significant deep learning progress for image recognition has raised strong research interest in medical image recognition. First of all, we found the prediction result using the VGG16 model on failed pneumonia X-ray images. Thus, this paper proposes IVGG13 (Improved Visual Geometry Group-13), a modified VGG16 model for classification pneumonia X-rays images. Open-source thoracic X-ray images acquired from the Kaggle platform were employed for pneumonia recognition, but only a few data were obtained, and datasets were unbalanced after classification, either of which can result in extremely poor recognition from trained neural network models. Therefore, we applied augmentation pre-processing to compensate for low data volume and poorly balanced datasets. The original datasets without data augmentation were trained using the proposed and some well-known convolutional neural networks, such as LeNet AlexNet, GoogLeNet and VGG16. In the experimental results, the recognition rates and other evaluation criteria, such as precision, recall and f-measure, were evaluated for each model. This process was repeated for augmented and balanced datasets, with greatly improved metrics such as precision, recall and F1-measure. The proposed IVGG13 model produced superior outcomes with the F1-measure compared with the current best practice convolutional neural networks for medical image recognition, confirming data augmentation effectively improved model accuracy.

**Keywords:** thoracic X-ray; deep learning; data augmentation; convolutional neural network; LeNet; AlexNet; GoogLeNet; VGGNet; Keras



## 1. Introduction

Most recent deep learning breakthroughs are related to convolutional neural networks (CNNs), which are also the main developing area for deep neural networks (DNNs). Modern CNN approaches can be more accurate than humans for image recognition. The ImageNet Large-Scale Visual Recognition Challenge (ILSVRC) [1] is a highly representative academic competition for machine vision solutions based on image data provided by ImageNet. The main dataset comprises more than 14 million marked images, with a smaller subset sampled for the yearly ILSVRC. The best ILSVRC result prior to 2012 achieved a 26% error rate. However, a CNN model based on AlexNet [2] reduced the error rate to 16.4% in 2012, winning the championship. Subsequent studies have used various CNN approaches, some going on to become major and well-known CNN architectures.

Many current products use deep learning technologies, most of which relate to image recognition. Recent hardware breakthroughs and significant improvements have brought a strong focus onto deep learning, with CNN models becoming the most popular approaches for image recognition, image segmentation [3–5] and object recognition. Image recognition refers to the process where a machine is trained using CNNs to extract important features

from large image datasets, combine them into a feature map and perform recognition by connecting neurons. The approach has been successfully applied to various areas, such as handwriting recognition [6–8], face recognition [9–11], automatic driving vehicles [12], video surveillance [13] and medical image recognition [14–17].

With the development of computer technology applied to the medical field, whether through basic medical applications, disease treatments, clinical trials or new drug therapies, all such applications involve data acquisition, management and analysis. Therefore, determining how modern medical information can be used to provide the required data is an important key to modern medical research. Medical services mainly include telemedicine, information provided through internet applications and digitization of medical information. In this way, we more accurately and quickly confirm a patient's physical condition and determine how best to treat a patient, thereby improving the quality of medical care. Smart healthcare can help us to establish an effective clinical decision support system to improve work efficiency and the quality of diagnosis and treatment. This is of particular importance in the aging population in society, which has many medical problems. In addition, the COVID-19 [18–22] outbreak in 2020 greatly increased the demand for medical information processing. Therefore, medical information is combined with quantitative medical research; in addition, assistance in diagnosis from an objective perspective is a trend at present. The development of big data systems has also enabled the systematic acquisition and integration of medical images [23–27]. Furthermore, traditional image processing algorithms have gradually been replaced by deep learning algorithms.

Recent deep learning and machine learning developments mean that traditional image processing method performance for image recognition is no longer comparable to that of neural network (NN)-based approaches. Consequently, many studies have proposed optimized deep learning algorithms to improve image recognition accuracy for various recognition scenarios. CNN is the most prominent approach for image recognition, improving recognition accuracy by increasing hidden layer depths and effectively acquiring more characteristic parameters. Successful image recognition applications include face, object, and license plate recognition, but medical image recognition is less common due to difficulties acquiring medical images and poor understanding regarding how diseases appear in the various images. Therefore, physician assistance is usually essential for medical image recognition to diagnose and label focal areas or lesions before proceeding to model training. This study used open-source thoracic X-ray images from the Kaggle data science community, which were already categorized and labeled by professional physicians. Recognition systems were pre-trained using LeNet [28], AlexNet [2], GoogLeNet [29] and VGG16 [30] images, but trained VGG16 model classification exhibited poor image classification accuracy in the test results. Therefore, this paper proposes IVGG13 to solve the problem of applying VGG16 to medical image recognition. Several other well-known CNNs were also trained on the same datasets, and the outcomes were compared with the proposed IVGG13 approach. The proposed IVGG13 model outperformed all other CNN models considered. We also applied data augmentation to increase the raw dataset and improve the data balance, hence improving the model recognition rate. It is essential to consider hardware requirements for CNN training and deployment. The number of network control parameters increases rapidly with increasing network layer depth, which imposes higher requirements on hardware and increases overhead and computing costs. Therefore, this paper investigated methods to reduce network depth and parameter count without affecting recognition accuracy. The proposed IVGG13 incorporates these learnings, and it has strong potential for practical medical image recognition systems.

The remainder of this paper is organized as follows. The related works are described in Section 2. The research methodology is stated in Section 3. The performance evaluation is outlined in Section 4. Finally, conclusions and suggestions for future research are provided in Section 5.

## 2. Materials and Methods

### 2.1. Deep Learning Model

The deep learning concept originated in 2006 and subsequently attracted sustained strong research and industrial interest. Many world-leading artificial intelligence (AI) companies have participated in the ILSVRC competition since its inception. Initial competition occurred between machine learning and support vector machine technologies. However, the first deep learning framework that won the championship in 2012 triggered rapid deep learning developments and the emergence of now well-known deep learning models, such as AlexNet [2], VGGNet [30] and GoogLeNet [29]. The following sections provide a brief introduction.

#### 2.1.1. LeNet

Figure 1 exhibits the LeNet architecture, also known as LeNet5 [28], which was the first CNN architecture for deep learning. LeNet was originally proposed by LeCun [28], who subsequently helped develop the more general CNN approaches and initially used it to recognize handwritten characters. Early models did not employ GPUs for training and were restricted to CPUs alone; hence, training was very slow compared with modern systems. The LeNet design incorporated convolutions, pooling and parameter sharing to extract features and effectively reduce computational overheads and complete classification and recognition through fully connected layers.

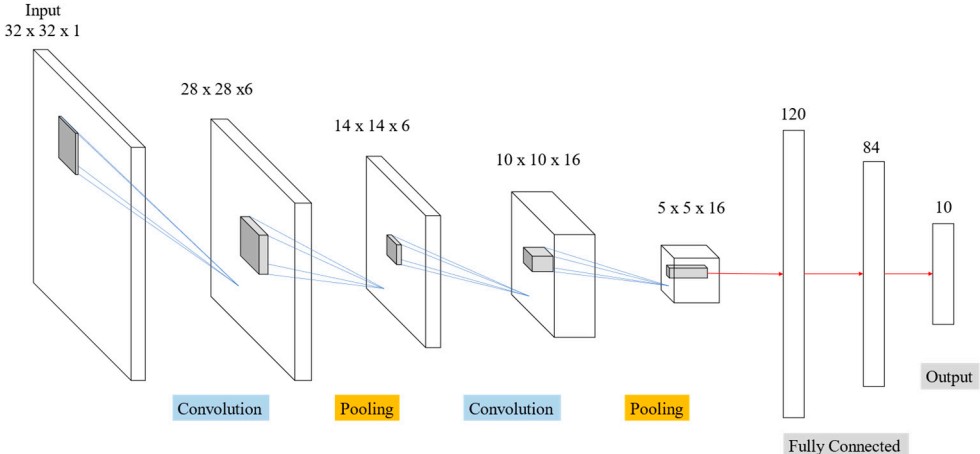

**Figure 1.** LeNet network architecture [28].

#### 2.1.2. AlexNet

AlexNet was the first CNN used in the LSVRC competition, proposed by Krizhevsky et al. 2012 [2], which won the competition with significantly improved accuracy compared with all previous models, including the one that took second place that year. AlexNet has three main features:

(1) Employs the ReLU non-linear activation function to solve the vanishing gradient problem more effectively than sigmoid and tanh activation functions used in other NNs;
(2) Adds dropout and data augmentation in the network layer to prevent overfitting; and
(3) Employs multiple parallel GPUs to accelerate computational throughput during training.

As shown in Figure 2, AlexNet architecture is similar to LeNet. The network architecture is divided into two layers since training is done on two GPUs due to memory restrictions, and data dropout or augmentation is added to prevent overfitting.

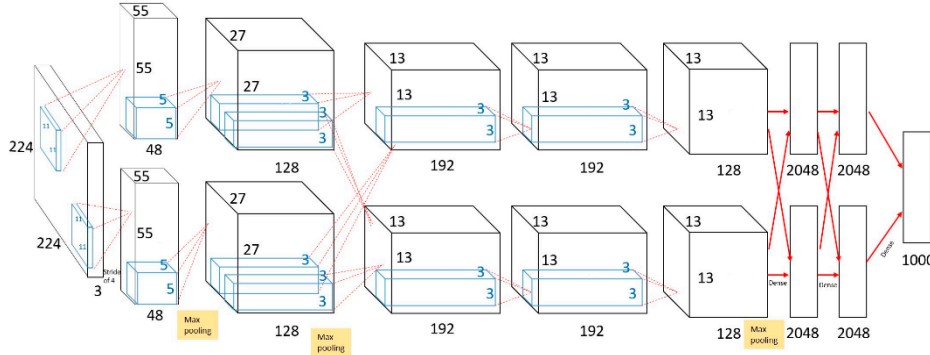

**Figure 2.** AlexNet network architecture [2].

### 2.1.3. VGGNet

VGGNet is a CNN jointly developed by the Visual Geometry Group at the University of Oxford and Google DeepMind [30]. As shown in Figure 3, VGGNet architecture can be considered an extended AlexNet, characterized by $3 \times 3$ convolutional kernels and $2 \times 2$ pooling layers, and the network architecture can be deepened by using smaller convolutional layers to enhance feature learning. The two most common current VGGNet versions are VGGNet-16 and VGGNet-19 [30].

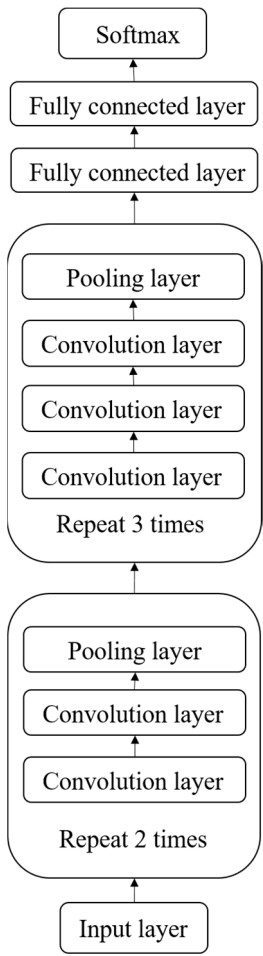

**Figure 3.** VGGNet network architecture.

### 2.1.4. GoogLeNet

The earliest GoogLeNet version, Inception V1, won the ILSVRC competition with higher accuracy than VGGNet in 2014 [29]. Figure 4 displays a typical GoogLeNet ar-

chitecture. Inception architecture was subsequently derived to deepen and widen the network by using receptive vision fields with different convolutional kernel sizes to improve network accuracy.

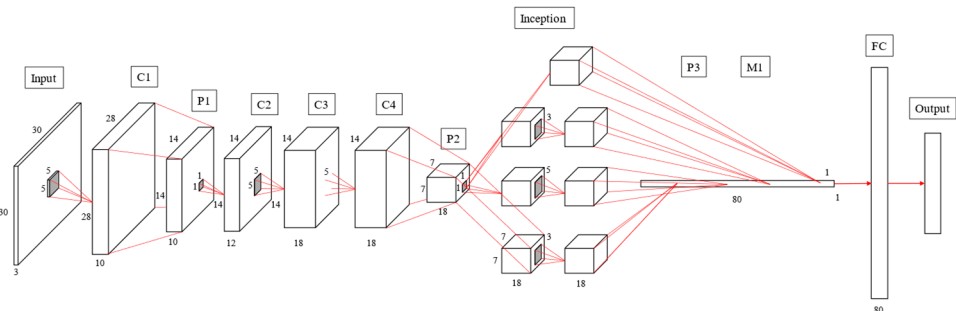

**Figure 4.** Typical GoogLeNet network architecture [29].

Figure 5 demonstrates that inception architecture contains convolutional $1 \times 1$, $3 \times 3$, and $5 \times 5$ kernels with maximum pooling $3 \times 3$ stacking. Different convolutional kernels sizes are used for feature extraction and connection to increase network width and enhance adaptability to different sizes. The $3 \times 3$ and $5 \times 5$ convolutional kernels are preceded by $1 \times 1$ convolutional kernels for dimensionality and parameter size reduction, reducing computing volume and correcting nonlinear functions. Finally, a $1 \times 1$ convolution is added after $3 \times 3$ maximum pooling.

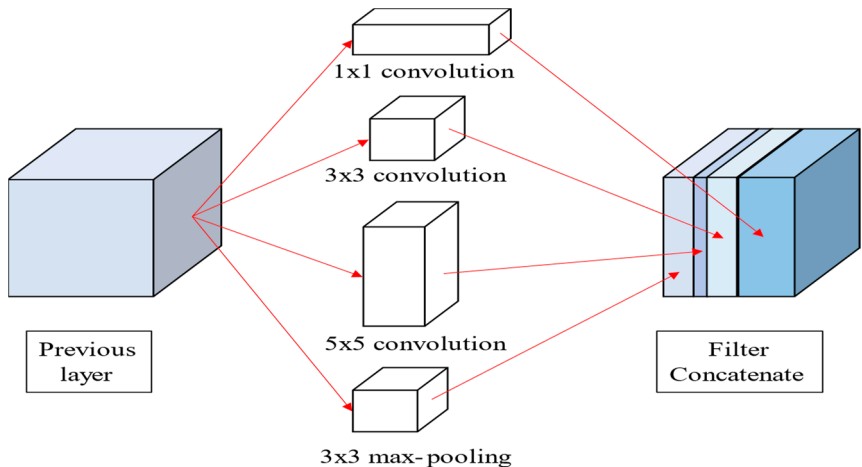

**Figure 5.** Typical inception module architecture [29].

## 3. Research Methods

This section discusses various study approaches. Section 3.1 introduces the source and classification for image datasets used in this study; Section 3.2 describes data augmentation pre-processing to solve unbalanced or small dataset problems; and Section 3.3 presents the proposed IVGG13 model.

Figure 6 shows the flow chart of data pre-processing and Figure 7 exhibits the proposed CNN training process. LeNet, AlexNet, GoogLeNet, VGG16 and IVGG13 models were trained with the original datasets without data augmentation and then evaluated and compared. The models were subsequently trained with augmented datasets and again evaluated and compared.

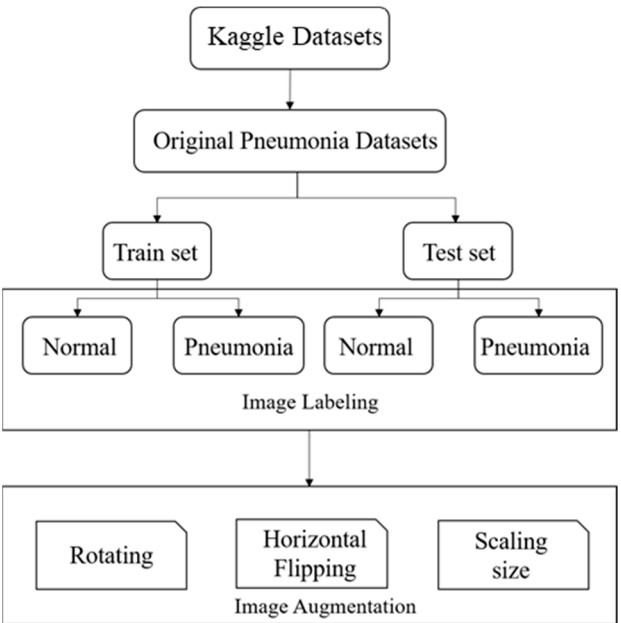

**Figure 6.** Data pre-processing.

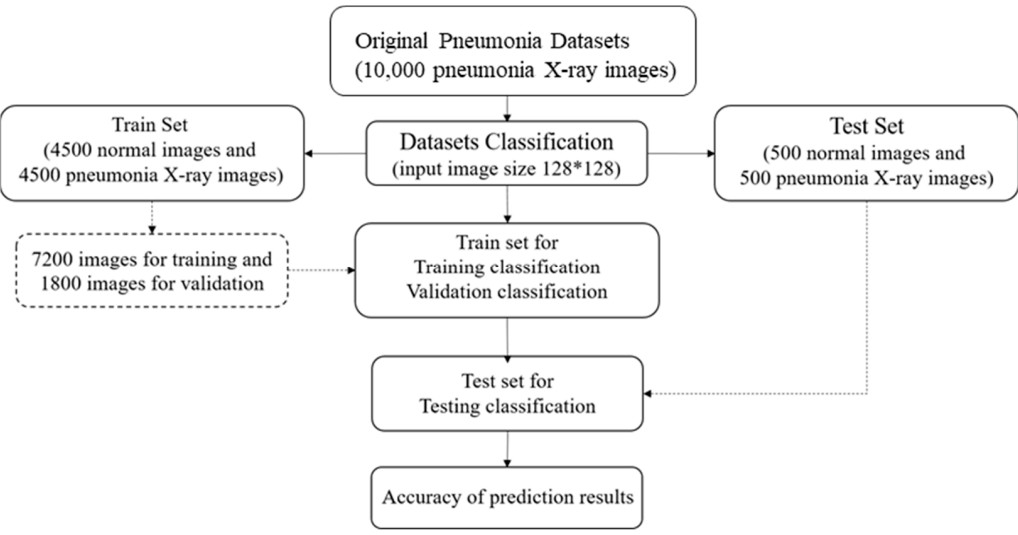

**Figure 7.** CNN network training.

### 3.1. Training Datasets

We used an open-source dataset provided by the Kaggle data science competition platform for training (https://www.kaggle.com/paultimothymooney/chest-xray-pneumonia accessed on 25 March 2018) [31]. The dataset comprised thoracic cavity images from child patients (1 to 5 years old) from the Guangzhou Women and Children's Medical Center, China. These images were classified by two expert physicians and separated into training, test and validation sets. Figure 8 displays the dataset structure, with training sets including 1341 and 3875, test sets 234 and 390, validation set 8, and eight normal and pneumonia images, respectively. Figures 9 and 10 show examples of normal and pneumonia thoracic cavity X-ray images, respectively.

```
∨  📁 chest_xray                    3 directories
  ∨  📁 test                       2 directories
       📁 NORMAL                     234 files
       📁 PNEUMONIA                  390 files
  ∨  📁 train                      3 directories
       📁 NORMAL                    1341 files
       📁 PNEUMONIA                 3875 files
  ∨  📁 val                        3 directories
       📁 NORMAL                       8 files
       📁 PNEUMONIA                    8 files
```

**Figure 8.** Training datasets for this study.

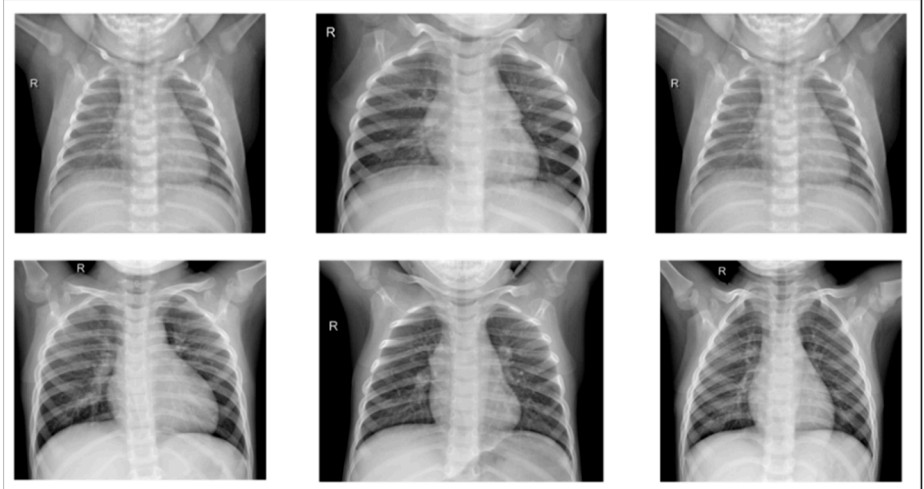

**Figure 9.** Example normal thoracic cavity X-ray images from the study dataset [31].

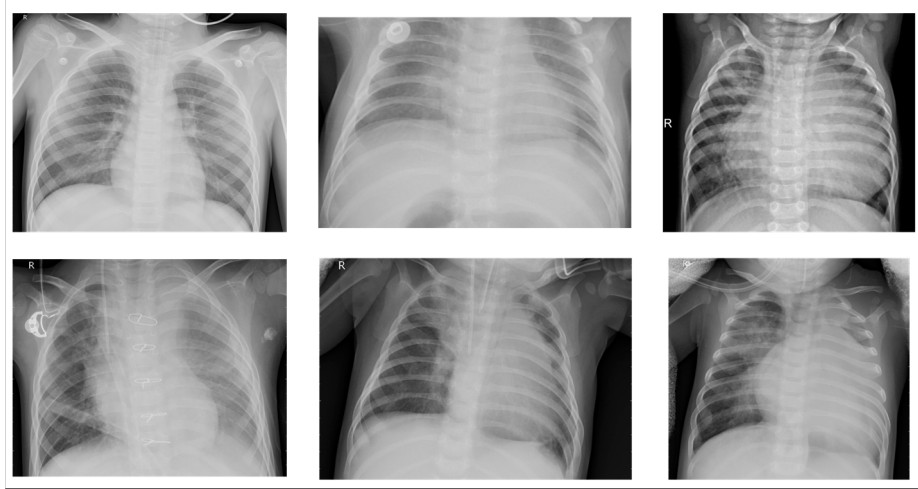

**Figure 10.** Example pneumonia thoracic cavity X-ray images from the study dataset [31].

### 3.2. Data Augmentation

The study dataset included unbalanced positive and negative samples, with significantly fewer normal images than pneumonia images in both training and test sets and relatively low data volume. This could lead to poor post-training validation and overfitting. Therefore, we applied data augmentation on the original datasets, creating new images by horizontal flipping, rotating, scaling size and ratio, and changing brightness and color

temperature for the original images to compensate for the lack of data volume. Data augmentation increased the training set from 5216 to 22,146 images, and the test set from 624 to 1000 images. Furthermore, some images were transferred from the training to test set for data balance and to ensure images in the test set were predominantly originals. Figures 11 and 12 show examples of original and augmented images, respectively.

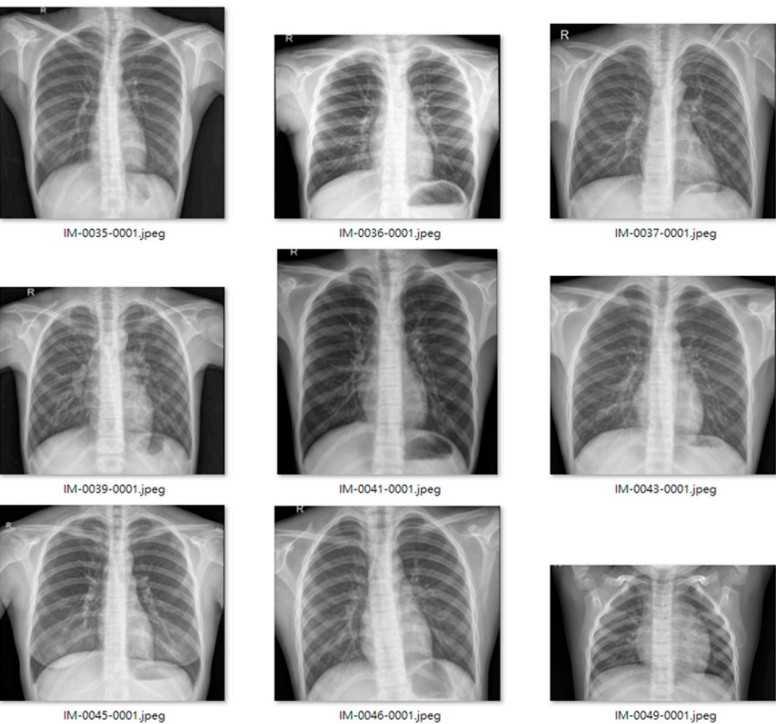

**Figure 11.** Example original images [31].

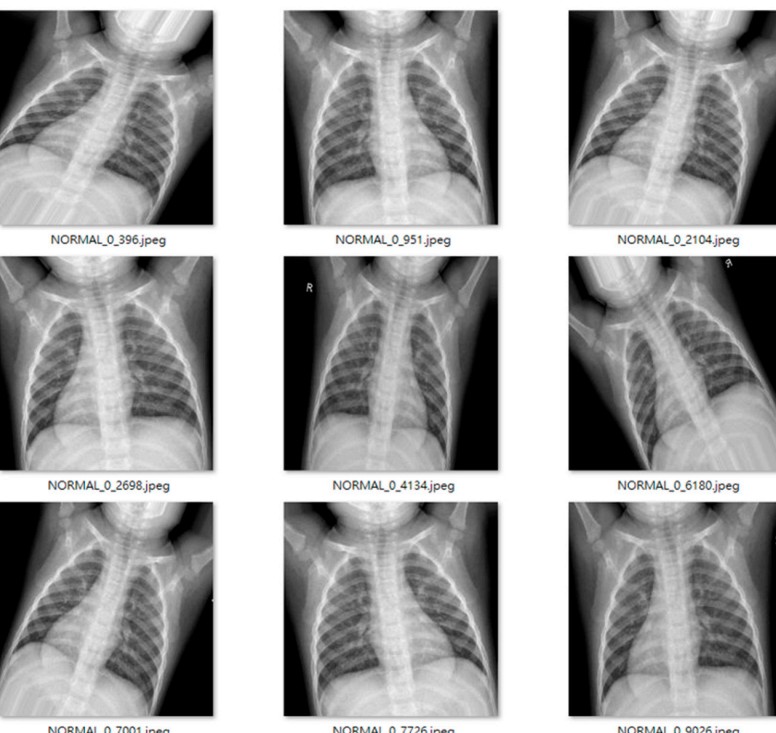

**Figure 12.** Example images after data augmentation.

### 3.3. IVGG13

This study proposes IVGG13—an improved VGG16 that reduces the VGGNet network depth—as shown in both Table 1 and Figure 13. The proposed network architecture reduces the number of parameters by reducing the network depth compared to the original VGG16 to avoid both under- and overfitting problems during training. The original VGG16 convolutional architecture was retained by performing feature extraction using two consecutive small convolutional kernels rather than a single large one. This maintains VGG16 perceptual effects while reducing the number of parameters, which not only reduces the training time but also maintains the network layer depth.

**Table 1.** Proposed IVGG13 network model.

| Layer (Type) | Output Shape | Param |
|---|---|---|
| conv2d_1 (conv2D) | (None, 128, 128, 32) | 896 |
| conv2d_2 (conv2D) | (None, 128, 128, 32) | 9248 |
| max_Pooling2d_1 (MaxPooling2) | (None, 64, 64, 32) | 0 |
| conv2d_3 (conv2D) | (None, 64, 64, 32) | 9248 |
| conv2d_4 (conv2D) | (None, 64, 64, 32) | 9248 |
| max_Pooling2d_2 (MaxPooling2) | (None, 32, 32, 32) | 0 |
| conv2d_5 (conv2D) | (None, 32, 32, 64) | 18,496 |
| conv2d_6 (conv2D) | (None, 32, 32, 64) | 36,928 |
| max_Pooling2d_3 (MaxPooling2) | (None, 16, 16, 64) | 0 |
| conv2d_7 (conv2D) | (None, 16, 16, 128) | 73,856 |
| conv2d_8 (conv2D) | (None, 16, 16, 128) | 147,584 |
| max_Pooling2d_4 (MaxPooling2) | (None, 8, 8, 128) | 0 |
| conv2d_9 (conv2D) | (None, 8, 8, 64) | 73,792 |
| conv2d_10 (conv2D) | (None, 8, 8, 64) | 36,928 |
| max_Pooling2d_5 (MaxPooling2) | (None, 4, 4, 64) | 0 |
| flatten_1 (Flatten) | (None, 1024) | 0 |
| dense_1 (Dense) | (None, 1024) | 1,049,600 |
| dropout_1 (Dropout) | (None, 1024) | 0 |
| dense_2 (Dense) | (None, 1024) | 1,049,600 |
| dropout_2 (Dropout) | (None, 1024) | 0 |
| dense_3 (Dense) | (None, 2) | 2050 |
| Total params: 2,517,474 | | |
| Trainable params: 2,517,474 | | |
| Non-trainable params: 0 | | |

Figure 13 highlights the similarities and differences between the IVGG13 and VGG16 network architectures.

First, the input image size was changed to 128 × 128, and the hidden layer was divided into five blocks, with each block containing two convolutional layers and a pooling layer. Thirty-two 3 × 3 convolutional kernels were randomly generated in each convolutional layer for feature extraction, and the image size was reduced by the pooling layer. Convolutional kernels in blocks 3–5 were the same size (3 × 3), but 64, 128 and 64 kernels in each block, respectively. Reducing convolutional kernels reduced the number of parameters required compared with VGG16. The image size was then reduced by the pooling layer, feature maps were converted to one dimension by the flattened layer and finally, three of the fully connected layers concatenated output features into two classifications.

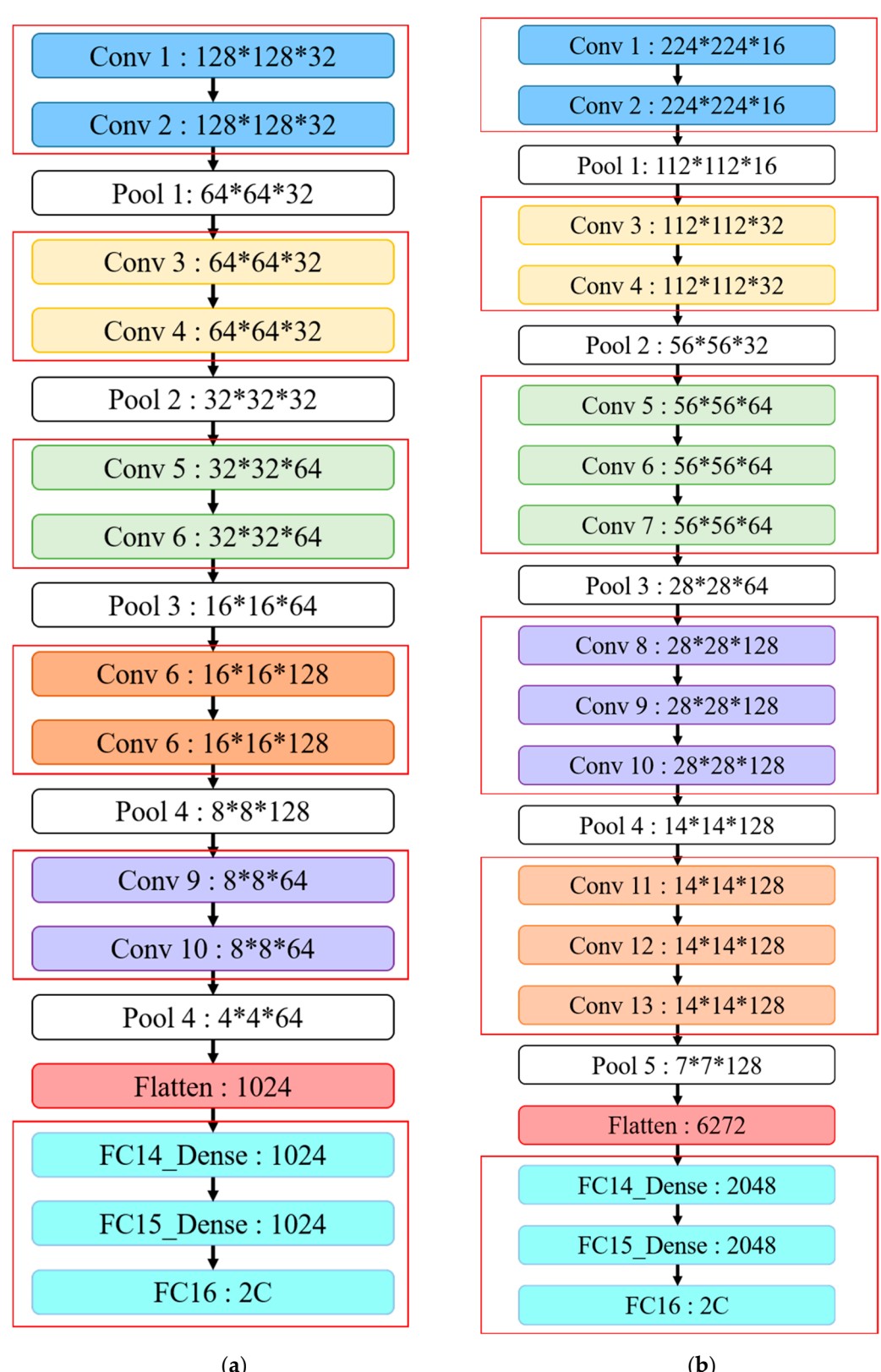

**Figure 13.** (**a**) Proposed IVGG13 and (**b**) conventional VGG16 [30] network architectures.

## 4. Results

This section provides a concise and precise description of the experimental results, their interpretation, as well as the experimental conclusions that can be drawn. Section 4.1 introduces the experimental environment of this article. Section 4.2 introduces and com-

pares various CNNs; Sections 4.3 and 4.4 discuss model outcomes without and with data pre-processing, respectively; and Section 4.5 discusses VGG16 problems highlighted by the experimental results.

### 4.1. Experimental Environment

This study used a workstation with Windows 10, Intel Core i5-8500 @ 3.00 GHz CPU, Nvidia GeForce RTX2070 8 G GPU, and 32.0 GB RAM. TensorFlow-GPU was employed to train the CNN in Python 3.5.6 by Anaconda3, with Python Keras to build the network architecture and training.

### 4.2. CNN Comparison

#### 4.2.1. LeNet

Table 2 and Figure 14 display outcomes from LeNet network for MNIST handwriting character recognition applied to train thoracic X-ray images. Sixteen $5 \times 5$ convolution kernels were randomly generated from each $28 \times 28$ input image, and the first convolution generated 16 ($28 \times 28$) images. Images then reduced to $14 \times 14$ using reduction sampling in the pooling layer. The second convolution converted the 16 images into 36 $14 \times 14$ images using $5 \times 5$ convolutional kernels. The image size was then further reduced to $7 \times 7$ by reduction sampling in the pooling layer. Finally, the features were converted into one dimension in the flattening layer, fully connected, and output as two categories.

**Table 2.** Network model for the LeNet implementation [28].

| Layer (Type) | Output Shape | Param |
|---|---|---|
| conv2d_1 (conv2D) | (None, 28, 28, 16) | 1216 |
| max_Pooling2d_1 (MaxPooling2) | (None, 14, 14, 16) | 0 |
| conv2d_2 (conv2D) | (None, 14, 14, 36) | 14,436 |
| max_Pooling2d_2 (MaxPooling2) | (None, 7, 7, 36) | 0 |
| flatten_1 (Flatten) | (None, 1764) | 0 |
| dense_1 (Dense) | (None, 128) | 225,920 |
| dense_2 (Dense) | (None, 2) | 258 |
| Total params: 241,830 | | |
| Trainable params: 241,830 | | |
| Non-trainable params: 0 | | |

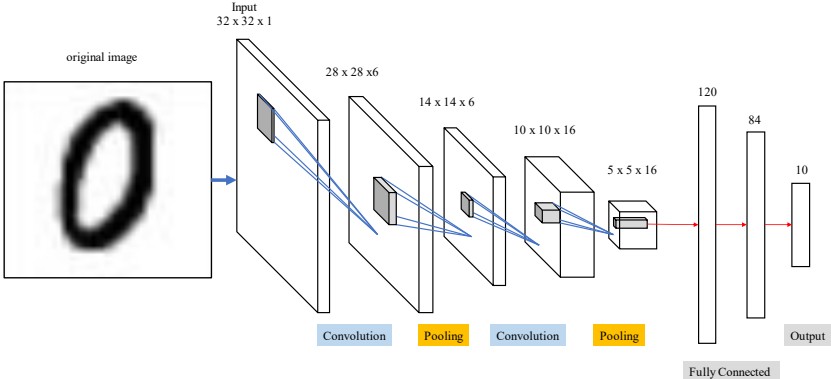

**Figure 14.** Network architecture for the LeNet MNIST handwriting character recognition implementation [28].

#### 4.2.2. AlexNet

Table 3 and Figure 15 exhibit the AlexNet architecture used to train thoracic X-ray images, which comprise five convolutional and three fully connected layers, with $227 \times 227$ input images. The first convolutional layer included 48 ($11 \times 11$) convolutional kernels to produce $227 \times 227$ images, followed by local response normalization in the LRN layer to reduce images to $55 \times 55$ using $3 \times 3$ max pooling. The second convolutional layer was

similar to the first, but included 128 (5 × 5) convolution kernels. Subsequent LRN and max pooling layers reduced image size to 13 × 13, with convolution layers 3–5 employing 192, 192 and 128 (3 × 3) kernels, respectively, producing 13 × 13 images. Images were reduced to 6 × 6 using a max pooling layer, and features were converted to one dimension in the flattening layer, fully connected, and output as two categories.

**Table 3.** AlexNet network model [2].

| Layer (Type) | Output Shape | Param |
|---|---|---|
| conv2d_1 (conv2D) | (None, 55, 55, 48) | 17,472 |
| max_Pooling2d_1 (MaxPooling2) | (None, 27, 27, 48) | 0 |
| conv2d_2 (conv2D) | (None, 27, 27, 128) | 153,728 |
| max_Pooling2d_2 (MaxPooling2) | (None, 13, 13, 128) | 0 |
| conv2d_3 (conv2D) | (None, 13, 13, 192) | 221,376 |
| conv2d_4 (conv2D) | (None, 13, 13, 192) | 331,968 |
| conv2d_5 (conv2D) | (None, 13, 13, 192) | 221,312 |
| max_Pooling2d_3 (MaxPooling2) | (None, 6, 6, 128) | 0 |
| flatten_1 (Flatten) | (None, 4608) | 0 |
| dense_1 (Dense) | (None, 2048) | 9,439,232 |
| dropout_1(Dropout) | (None, 2048) | 0 |
| dense_2 (Dense) | (None, 2048) | 4,196,352 |
| dropout_2(Dropout) | (None, 2048) | 0 |
| dense_3 (Dense) | (None, 2) | 4098 |
| Total params: 14,585,538 | | |
| Trainable params: 14,585,538 | | |
| Non-trainable params: 0 | | |

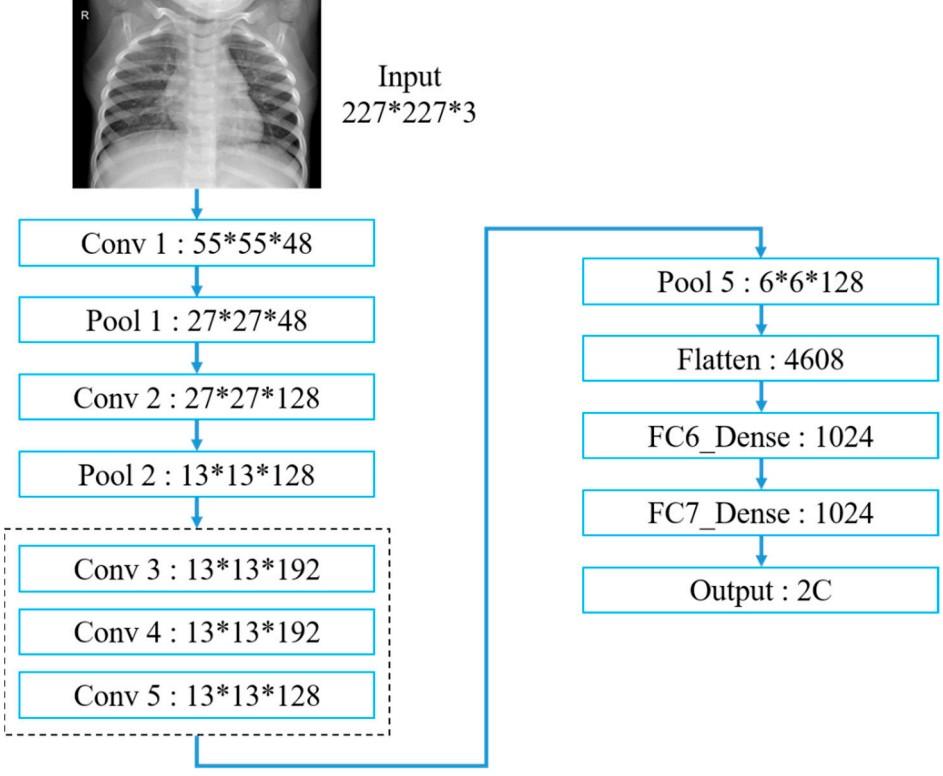

**Figure 15.** AlexNet architecture employed [2].

### 4.2.3. VGG16

Considering the selected workstation performance, we only used VGG16 CNN for this study. First, we used the VGG16 model and applied it to pneumonia X-ray data for training, but the prediction results failed. Table 4 and Figure 16 indicate that our VGG16

architecture contains 13 convolutional and 3 fully connected layers, with 3 × 3 kernels for the convolutional layers and 2 × 2 parameters for the pooling layers. VGG16 convolutional and pooling layers are divided into blocks 1–5, where each block contains multiple convolutional layers and a single pooling layer. The two convolutional layers in block 1 each use 16 kernels for feature extraction, with image size subsequently reduced in the pooling layer. Subsequent blocks have similar architecture, except that blocks 1 and 2 use two convolutional layers, whereas blocks 3–5 use three convolutional layers with different kernel numbers in each layer to deepen the network and improve accuracy. Finally, three fully connected layers concatenate and output features into two classifications.

**Table 4.** VGG16 network model [30].

| Layer (Type) | Output Shape | Param |
|---|---|---|
| conv2d_1 (conv2D) | (None, 224, 224, 16) | 448 |
| conv2d_2 (conv2D) | (None, 224, 224, 16) | 2320 |
| max_Pooling2d_1 (MaxPooling2) | (None, 112, 112, 16) | 0 |
| conv2d_3 (conv2D) | (None, 112, 112, 32) | 4640 |
| conv2d_4 (conv2D) | (None, 112, 112, 32) | 9248 |
| max_Pooling2d_2 (MaxPooling2) | (None, 56, 56, 32) | 0 |
| conv2d_5 (conv2D) | (None, 56, 56, 64) | 18,496 |
| conv2d_6 (conv2D) | (None, 56, 56, 64) | 36,928 |
| conv2d_7 (conv2D) | (None, 56, 56, 64) | 36,928 |
| max_Pooling2d_3 (MaxPooling2) | (None, 28, 28, 64) | 0 |
| conv2d_8 (conv2D) | (None, 28, 28, 128) | 73,856 |
| conv2d_9 (conv2D) | (None, 28, 28, 128) | 147,584 |
| conv2d_10 (conv2D) | (None, 28, 28, 128) | 147,584 |
| max_Pooling2d_4 (MaxPooling2) | (None, 14, 14, 128) | 0 |
| conv2d_11 (conv2D) | (None, 14, 14, 128) | 147,584 |
| conv2d_12 (conv2D) | (None, 14, 14, 128) | 147,584 |
| conv2d_13 (conv2D) | (None, 14, 14, 128) | 147,584 |
| max_Pooling2d_5 (MaxPooling2) | (None, 7, 7, 128) | 0 |
| flatten_1 (Flatten) | (None, 6272) | 0 |
| dense_1 (Dense) | (None, 2048) | 12,847,104 |
| dropout_1(Dropout) | (None, 2048) | 0 |
| dense_2 (Dense) | (None, 2048) | 4,196,352 |
| dropout_2(Dropout) | (None, 2048) | 0 |
| dense_3 (Dense) | (None, 2) | 4098 |
| Total params: 17,968,338 | | |
| Trainable params: 17,968,338 | | |
| Non-trainable params: 0 | | |

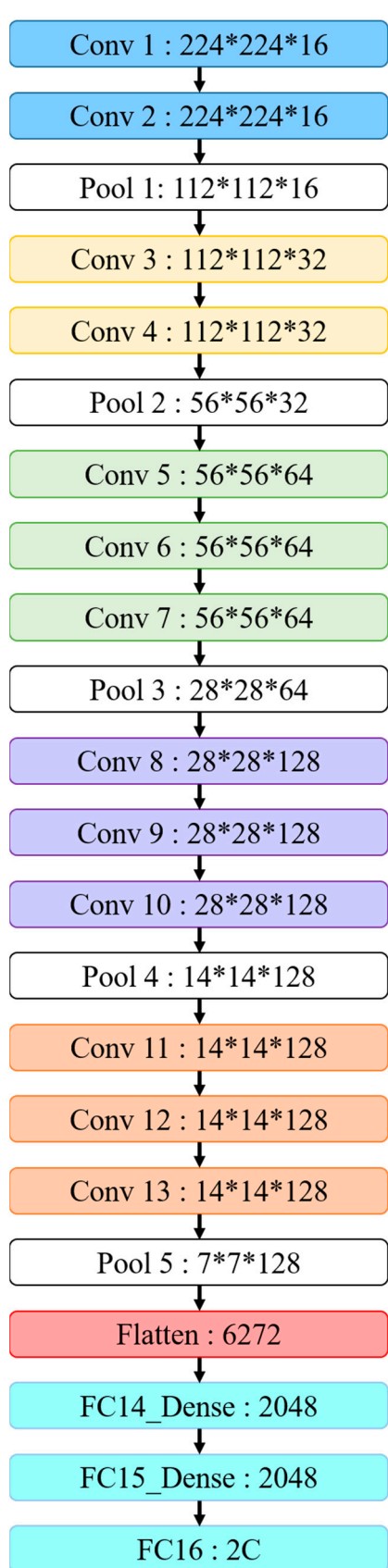

**Figure 16.** VGG16 network architecture [30].

### 4.3. Results without Data Pre-Processing

This section discusses the recognition rates for LeNet, AlexNet, VGG16, GoogLeNet and IVGG13 trained without data augmentation. The dataset before augmentation comprised 1349 and 3883, and 234 and 390 normal and pneumonia images in the training and test sets, respectively. The training parameters included the learning rate = 0.001, maximum epoch = 60 and batch size = 64.

Table 5 display the average confusion matrix outcomes over five repeated trainings for each network, respectively.

**Table 5.** Each model confusion matrix on the test set.

| Confusion Matrix on the Test Set | | | | |
|---|---|---|---|---|
| Model Name | TP | FP | TN | FN |
| LeNet | 387 | 143 | 91 | 3 |
| AlexNet | 386 | 160 | 74 | 4 |
| GoogLeNet | 306 | 82 | 152 | 84 |
| IVGG13 | 388 | 138 | 96 | 2 |

Figure 7 compares the overall network performance calculated using Equations (1)–(4). A good prediction model requires not only high accuracy but also generalizability. Generally, the validation is initially high, because the validation data is primarily used to select and modify the model; if the right validation data is selected at the beginning, the value of validation will exceed the accuracy value of the training set. Conversely, if the wrong data is selected, the parameters will be corrected and updated.

Evaluation metrics are usually derived from the confusion matrix (Tables 6 and 7) to evaluate classification results, where true positive (TP) means both actual and predicted results are pneumonia; true negative (TN) means both actual and predicted results are normal; false positive (FP) means actual results are normal but predicted to be pneumonia; and false negative (FN) means actual results are pneumonia but predicted to be normal.

**Table 6.** General confusion matrix.

| | **Actual YES** | **Actual NO** |
|---|---|---|
| Predicted YES | TP | FP |
| Predicted NO | FN | TN |

**Table 7.** Confusion matrix for medical judgment.

| | | **Actual Category** | |
|---|---|---|---|
| | | **Pneumonia** | **Normal** |
| Predicted category | Pneumonia | True positive (TP) | False positive (FP) |
| | Normal | False negative (FN) | True negative (TN) |

Table 8 compares the evaluation results for each model, where accuracy, precision, recall and F-measure were calculated as shown in Equations (1)–(4). Precision is the proportion of relevant instances among those retrieved, recall is the proportion of relevant instances that were retrieved and the F1-measure is a special case of the F-measure for $\beta = 1$, i.e., precision and recall are equally important. A larger F1 means improved model effectiveness.

$$\text{Accuracy} = \frac{\text{TP} + \text{TN}}{\text{TP} + \text{FP} + \text{FN} + \text{TN}} \tag{1}$$

$$\text{Precision} = \frac{\text{TP}}{\text{TP} + \text{FP}} \tag{2}$$

$$\text{Recall} = \frac{\text{TP}}{\text{TP} + \text{FN}} \tag{3}$$

$$\mathrm{F}_\beta - \mathrm{Measure} = \left(1+\beta^2\right)\frac{\mathrm{Precision} \times \mathrm{Recall}}{\left(\beta^2 \times \mathrm{Precision}\right) + \mathrm{Recall}} \tag{4}$$

**Table 8.** Prediction results for each network model.

|  | LeNet | AlexNet | GoogLeNet | IVGG13 |
|---|---|---|---|---|
| Training time | 49 s | 284 s | 1270 s | 231 s |
| Accuracy | 76.6% | 73.6% | 73.3% | **77.5%** |
| Precision | 73.0% | 70.6% | 78.8% | **73.7%** |
| Recall | 99.2% | 98.9% | 78.4% | **99.4%** |
| F1-Measure | 84.1% | 82.4% | 78.6% | **84.6%** |

### 4.4. Results after Data Pre-Processing

We applied data augmentation to investigate the effects on model accuracy using datasets containing 4000, 5000, 6000, 7000, 8000, 9000 and 10,000 randomly selected images from the training set for each network model. Training parameters included learning rate = 0.001, maximum epoch = 60 and batch size = 64.

Table 9 exhibits performance metrics calculated using Equations (1)–(4) for comparison by using the different datasets. Table 10 compares the best evaluations for each model. Although GoogleNet has higher accuracy and precision, the recall rate is much lower. It is therefore important to use the F1-measure as the model evaluation standard. Both precision and recall are equally important.

**Table 9.** Prediction results for IVGG13 model.

| Dataset | 4000 | 5000 | 6000 | 7000 | 8000 | 9000 | 10,000 |
|---|---|---|---|---|---|---|---|
| Training time | 188 s | 231 s | 279 s | 336 s | 375 s | 448 s | 476 s |
| Accuracy | 87.7% | 88.4% | **89.1%** | 86.4% | 86.5% | 87.5% | 86.8% |
| Precision | 81.3% | 82.2% | **83.3%** | 79.4% | 79.5% | 80.8% | 79.5% |
| Recall | 97.8% | 98.0% | 97.8% | 98.2% | 98.4% | 98.2% | **99.0%** |
| F1-Measure | 88.8% | 89.4% | **90.0%** | 87.8% | 87.9% | 88.7% | 88.2% |

**Table 10.** Prediction results for each network model.

|  | LeNet | AlexNet | GoogLeNet | IVGG13 |
|---|---|---|---|---|
| Training time | 34 s | 346 s | 1995 s | 279 s |
| Accuracy | 86.8% | 86.6% | **89.5%** | 89.1% |
| Precision | 80.0% | 81.0% | **89.3%** | 83.3% |
| Recall | 98.1% | 95.6% | 89.6% | **97.8%** |
| F1-Measure | 88.1% | 87.7% | 89.4% | **90.0%** |

### 4.5. VGG16 Problems

The preceding analysis highlighted several VGG16 problems with extracting features from medical image datasets. VGG16 was originally applied in the ILSVRC to recognize 1000 categories from 1 million images. Therefore, applying it to small datasets with fewer training features led to significant underfitting [32]. It has always been difficult to distinguish pneumonia presence or absence on thoracic X-ray images; image features are too homogeneous, making it difficult for models to capture relevant features. Therefore, training with deeper network layers often creates recognition errors.

Table 11 confirms the matrix for each dataset and that VGG16 also failed, even after augmentation.

**Table 11.** VGG16 confusion matrix for each dataset.

|        | TP  | FP  | TN | FN |
|--------|-----|-----|----|----|
| 4000   | 500 | 500 | 0  | 0  |
| 5000   | 500 | 500 | 0  | 0  |
| 6000   | 500 | 500 | 0  | 0  |
| 7000   | 500 | 500 | 0  | 0  |
| 8000   | 500 | 500 | 0  | 0  |
| 9000   | 500 | 500 | 0  | 0  |
| 10,000 | 500 | 500 | 0  | 0  |

## 5. Conclusions and Prospects

### 5.1. Conclusions

Since LeNet's emergence, CNNs have continued progressing with many breakthroughs and developments, and they provide great benefits for image recognition. Computational hardware and capacity have also improved significantly, supporting DNN requirements and extending their applicability. Therefore, developing deeper NNs to extract features can effectively and continuously improve recognition accuracy and, hence, modern CNN architectures commonly include many layers. First of all, we discovered the prediction result of using the VGG16 model on failed pneumonia X-ray images. Thus, this paper proposes IVGG13 (Improved Visual Geometry Group-13), a modified VGG16 model for the classification pneumonia X-rays images. Therefore, this paper proposed IVGG13, a modified CNN for medical image recognition by using open-source thoracic X-ray images from the Kaggle platform for training. The iVGG13 recognition rate was compared with the best current practice CNNs, which confirmed IVGG13's superior performance in medical image recognition and also highlighted VGG16 problems.

Data augmentation was employed to effectively increase data volume and balance before training. Recognition accuracy without data augmentation ranged from 74% to 77%, increasing to >85% after data augmentation, which confirmed that data augmentation can effectively improve recognition accuracy.

The proposed IVGG13 model required less training time and resources compared with the other considered CNNs. IVGG13 reduced layer depth with smaller convolutional kernels and, hence, used significantly less network parameters; this greatly reduced hardware requirements while achieving comparable or superior recognition accuracy compared with the other models considered. An accuracy of ≈89% was achieved after training with the augmented dataset, which is significantly superior to the other current best practice models.

### 5.2. Future Research Directions

Outcomes from this study suggest the following future directions:

1.  We expect the proposed IVGG13 to improve its model recognition rate to above 90% after optimizing the network architecture or image processing methods, ensuring a high recognition rate and stable system with good performance for future practical application in medical clinical trials.
2.  The proposed IVGG13 model with enhanced performance in X-ray image recognition is expected to be used for the study of kidney stones issues, such as KUB images classification.
3.  The proposed IVGG13 model could be combined with object detection to effectively provide physicians with accurate focal area detection during diagnosis to facilitate early disease detection and prevention.

**Author Contributions:** Conceptualization and methodology, Z.-P.J., Y.-Y.L. and K.-W.H. software—Z.-P.J. and Z.-E.S. formal analysis—Y.-Y.L. writing—original draft preparation, Z.-P.J., Y.-Y.L., Z.-E.S. and K.-W.H.; writing—review and editing, Z.-P.J., Y.-Y.L., Z.-E.S. and K.-W.H. All authors have read and agreed to the published version of the manuscript.

**Funding:** This work was supported in part by the Ministry of Science and Technology, Taiwan, R.O.C., under grants MOST 110-2222-E-992-006-.

**Institutional Review Board Statement:** Not applicable.

**Informed Consent Statement:** Not applicable.

**Data Availability Statement:** The data presented in this study are available on request from the corresponding author.

**Conflicts of Interest:** The authors declare no conflict of interest.

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
