# Peer review of "An Improved VGG16 Model for Pneumonia Image Classification"

_applsci, doi:10.3390/app112311185_

Round 1

Reviewer 1 Report

Please see the attached document 

Author Response

Reviewer 1:

Reviewer Comment P 1.1—Your title states “An improved VGG16 model ...”. It is unclear whether the model is VGG16 or VGG13 since the text refers to proposed method as VGG13. Please make this clearer

Response: Thank you very much for pointing out this shortcoming. At first, we used the VGG16 model to apply to pneumonia X-ray data for training, but the prediction results failed. Therefore, we improved the VGG16 model through our methods, and obtained the weight layer from 10 convolutional layers, plus three fully connected layers. Therefore, we named it IVGG13.

Reviewer Comment P 1.2—The references are not included properly in the text. Refer to MDPI Applied Sciences format. The references are seen as numbers mixed into the writing

Response: Thank you very much for pointing out this shortcoming. Following your suggestion, we'll refer to MDPI Applied Sciences format to edit our references.

Reviewer Comment P 1.3—The introductory paragraph includes irrelevant material which does not connect with the topic of the paper. Material regarding the recognition challenge should not be included. Consider adding the recognition challenge only as a reference instead. More importance should be placed on the topic. Your article title is “An Improved VGG16 Model for Pneumonia Image Classification”. The introductory section should include information about pneumonia and examples/challenges of image classification of pneumonia (include references). Much of the information could be used for a review article on CNN’s instead

Response: Thank you very much for pointing out this shortcoming. Following your suggestion, we have added to the introductory section information about pneumonia and examples/challenges of image classification of pneumonia (including references). For further details, please refer to the revised version Section 1 of the paper.

Reviewer Comment P 1.4—Too many figures. Consider removing some and adding the rest as a supplemental file. Only use figures which highlight or compares your proposed method to others. Prediction results/ tables for every number of images is not needed.

Response: Thank you very much for pointing out this shortcoming. Following your suggestion, we have moved many unnecessary prediction results, tables, and figures to the supplemental file. For more details, please refer to the supplement file (appendix).

Reviewer Comment P 1.5—The text does not reference the figures. Instead “Error! Reference source not found” is shown

Response: Thank you very much for pointing out this shortcoming. This is an error from the file conversion. We have corrected it carefully throughout the whole article.

Reviewer Comment P 1.6—Provide more detail about the novelty of the VGGNet framework compared to previous versions and other CNN’s. Describe the VGGNet success in applications. Describe some drawbacks of the other CNNs. It is not clear why the authors chose the VGGNet framework

Response: Thank you very much for pointing out this shortcoming. First of all, this is our first research on the convolutional neural network. We tried to use different convolutional neural networks (such as LeNet, AlexNet, VGG16, and GoogleNet) to train pneumonia X-ray data. Only the prediction result of using the VGG16 model failed, so we decided to improve the VGG16 model to help solve the problem of its prediction failure on pneumonia X-rays.

Reviewer Comment P 1.7—The original article format text is not removed. See the Results section

Response: Thank you very much for pointing out this shortcoming. We have corrected it carefully throughout the whole article.

Reviewer Comment P 1.8—The authors conclude that the proposed model outperforms the other CNN’s with less training time. This is not true as seen in the prediction tables comparing the CNN performance. Explain the success of the LeNet CNN

Response: Thank you very much for pointing out this shortcoming. According to this question, LeNet’s training time is less than IVGG13. However, we look at this question from a different perspective. LeNet was proposed earlier than VGGNet and its weight layer has just three layers, but we proposed IVGG13's weight layer has 13 layers. The training time of the convolutional neural network is related to the network layer numbers of the model, in that deeper network layers have a long time to train. Nevertheless, our experiment is mainly to solve the problem of VGG16 prediction failure on pneumonia X-ray.

Reviewer Comment P 1.9—Many of the references in the paper are from conferences which mainly show the status of research projects at the time of the conference. It is best practice to include the journal publication which show finalized results.

Response: Thank you very much for pointing out this shortcoming. Following your suggestion, we have included more than ten newer journal publications as references.

Reviewer 2 Report

Review for the manuscript:

Entitled: "An Improved VGG16 Model for Pneumonia Image Classification"

for Applied Sciences.

With ID: applsci-1438170

Dear author,

Thank you for your manuscript.

General comments

In this work, the authors present a modified neural network (NN) adapted to feature recognition of medical X-ray images, that exhibits improved performance in comparison to other established NNs. The article is well within the scope of Applied Sciences and it may be of interest to most of the readers of this journal. It shows an introductory background material, sufficient for someone not an expert in this area to understand the context and significance of this work, with good references to follow. Furthermore, Turnitin report indicated 20% similarity index. However, my main concern emerges from the very first line of the abstract: ‘Image recognition has been applied to many fields but relatively rarely for medical images.’ With a fast review in the literature someone can found countless articles dealing with Image recognition such as convolutional neural network in medical imaging. Indeed, few results can be found for Inception-based VGG (IVGG) networks, however this should be clarified by the authors and provide more details regarding IVGGs. In the results section there are too many prediction results and accuracy convergence data, but no accurate justification and discussion upon these chaotic results. The manuscript is obviously prepared in a hurry, and suffers from dispersed grammatical/syntax errors, of minor importance though, as well as from broken reference links. Please see specific comments below. A proofread could be beneficial.

For all the above and the specific comments below, I have opted to recommend a Major Revision for the current version of the manuscript.

Specific comments

L14: ‘This paper proposes IVGG13’ Please define every abbreviation, the first time cited in the text.

L20: ‘were trained using well-known and the proposed convolutional neural networks’ Well-known what? Something is missing here. Please revise. Generally, L20-23 should be revised in order to improve readability.

L23: ‘with greatly improved metrics’ Such as? Please be specific and provide values.

L23: ‘The proposed IVGG13 model produced superior’ Superior in terms of what? Please define and be specific.

L68: ‘The proposed IVGG13 model outperformed all other CNN models considered.’ In what terms? Please define.

L 77-80: "Section 2" should become 2.1; 3 -> 2.2; 4 -> 3; 5 -> 4

L 87-89: Please check expression

Fig 2: the upper part is truncated

L 148: "discussed"

L 211-212: Please check expression

L 215-218: Text copied from the template

L 241-242: Please check expression

L 244: "produces"

L 245: "pooling of."

L 249: "Images was"

L257: ‘Considering the selected workstation performance, we only used VGG16 CNN for this study.’ Please provide more justification on this selection.

L 317: It would be towards completeness if authors could include a short explanation of every performance metric.

Regarding the References, the DOIs should also be included.

Figures 17-20 contain the X-ray images and the respective accuracy-epoch graphs, but in my opinion, the X-ray images simply take up specs and do not contribute anything to the understanding of the procedure. It would improve readability if the authors could retain only the graphs, ideally combining them into one (clear and well-sized) in order to facilitate comparison between network performances.

Tables 5-8 can also be combined into one, for the sake of comparison.

In the same way, Figures 21-27 could be combined, omitting the X-ray images, as well as the Tables 12-18.

The same applies for Figures 28-34 with Tables 19-25; Figures 35-41 with Tables 26-32; Figures 42-48 with Tables 33-39. Similarly for the Section 3.5.

Author Response

Reviewer 2:

Reviewer Comment P 2.1—Turnitin report indicated 20% similarity index

Response: We have improved the presentation of our manuscript and corrected it carefully throughout the whole article. Finally, the manuscript underwent native English language editing.

Reviewer Comment P 2.2—In the results section there are too many prediction results and accuracy convergence data, but no accurate justification and discussion upon these chaotic results.

Response: Thank you very much for pointing out this shortcoming. Based on this question, we added information about our unique challenges related to this task in the abstract. We first took LeNet, AlexNet, GoogLeNet, and VGG16 to train the identification model. Finally, through the training results that we find found that the VGG16 prediction model was failed to predict on medical images.

Reviewer Comment P 2.3—L14: ‘This paper proposes IVGG13’ Please define every abbreviation, the first time cited in the text.

Response: Thank you very much for pointing out this shortcoming. We named it IVGG13 to improve upon VGG16. VGG is the abbreviation of Visual Geometry Group, and thirteen is the weight layer from our convolutional neural network model. In the abstract section, we have defined the sentence of IVGG13 as “This paper proposes IVGG13 (Improved Visual Geometry Group-13), a modified convolutional neural network, for medical image recognition”.

Reviewer Comment P 2.4—L20: ‘were trained using well-known and the proposed convolutional neural networks’ Well-known what? Something is missing here. Please revise. Generally, L20-23 should be revised in order to improve readability.

Response: Thank you very much for pointing out this shortcoming. Following your suggestion, we have corrected it carefully throughout the whole article.

Reviewer Comment P 2.5—L23: ‘with greatly improved metrics’ Such as? Please be specific and provide values.

Response: Thank you very much for pointing out this shortcoming. Based on this question, we have corrected it to “This process was repeated for augmented and balanced datasets, with greatly improved metrics such as precision, recall and F1-measure.”

Reviewer Comment P 2.6—The proposed IVGG13 model produced superior’ Superior in terms of what? Please define and be specific.

Response: Thank you very much for pointing out this shortcoming. Based on this question, we proposed an improved convolutional neural network IVGG13 is used for medical image recognition. The open-source thoracic X-ray images acquired from the Kaggle platform were employed for training, and the recognition rate of the model evaluation is compared with each well-known convolutional neural network. Finally, the proposed IVGG13 model produced superior outcomes with F1-measure compared with cur-rent best practice convolutional neural networks for medical image recognition, confirming data augmentation effectively improved model accuracy.

Reviewer Comment P 2.7—L68: ‘The proposed IVGG13 model outperformed all other CNN models considered.’ In what terms? Please define.

Response: Thank you very much for pointing out this shortcoming. Following your suggestion, we have corrected it outperformed referred to F1-measure.

Reviewer Comment P 2.8—L 77-80: "Section 2" should become 2.1; 3 -> 2.2; 4 -> 3; 5 ->4.

Response: Thank you very much for pointing out this shortcoming. Following your suggestion. We have corrected the labeling carefully throughout whole article

Reviewer Comment P 2.9—L 87-89: Please check expression, Fig 2: the upper part is truncated, L 148: "discussed", L 211-212: Please check expression, L 215-218: Text copied from the template, L 241-242: Please check expression, L 244: "produces", L 245: "pooling of.", L 249: "Images was"

Response: Thank you very much for pointing out this shortcoming. Following your suggestion, we have corrected it carefully throughout the whole article.

Reviewer Comment P 2.10—L257: ‘Considering the selected workstation performance, we only used VGG16 CNN for this study.’ Please provide more justification on this selection.

Response: Thank you very much for pointing out this shortcoming. Based on this question, at first, we used the VGG16 model to apply to pneumonia X-ray data for training, but the prediction results failed. Therefore, we improved the VGG16 model through our methods to solve the problem of its prediction failure on pneumonia X-rays.

Reviewer Comment P 2.11—L 317: It would be towards completeness if authors could include a short explanation of every performance metric.

Response: Thank you very much for pointing out this shortcoming. Following your suggestion, we have given a brief explanation of factors of performance and comparison factors. Precision is the pro-portion of relevant instances among those retrieved, recall is the proportion of relevant in-stances that were retrieved, and F1-measure is a special case of F-measure for β = 1, i.e., precision and recall are equally important. Larger F1 means improved model effective-ness.

Reviewer Comment P 2.12—Regarding the References, the DOIs should also be included.

Response: Thank you very much for pointing out this shortcoming. Following your suggestion, we have added references including DOIs.

Reviewer Comment P 2.13—Figures 17-20 contain the X-ray images and the respective accuracy-epoch graphs, but in my opinion, the X-ray images simply take up specs and do not contribute anything to the understanding of the procedure. It would improve readability if the authors could retain only the graphs, ideally combining them into one (clear and well-sized) in order to facilitate comparison between network performances.

Response: Thank you very much for pointing out this shortcoming. Following your suggestion, we have moved many unnecessary prediction results, tables, and figures to the supplemental file. For more details, please refer to the supplement file of this article.

Reviewer Comment P 2.14—Tables 5-8 can also be combined into one, for the sake of comparison. In the same way, Figures 21-27 could be combined, omitting the X-ray images, as well as the Tables 12-18. The same applies for Figures 28-34 with Tables 19-25; Figures 35-41 with Tables 26-32; Figures 42-48 with Tables 33-39. Similarly for the Section 3.5.

Response: Thank you very much for pointing out this shortcoming. Following your suggestion, we have moved many unnecessary prediction results, tables, and figures to the supplemental file. For more details, please refer to the supplement file of this article.

Reviewer 3 Report

The work introduces a modified CNN for medical image recognition called IVGG13. For training, they used open-source thoracic X-Ray images from the Kaggle platform. In addition, the authors employed data augmentation to increase the data volume and balance before training.

Something went wrong with the references, either the text shows something like “image recognition3412131415” or it displays “Error! Reference source not found.”. So, it is hard to say, which reference belongs where in the text and if there are maybe even references missing. Please fix this first.

Author Response

Reviewer 3:

Reviewer Comment P 3.1—Something went wrong with the references, either the text shows something like “image recognition3412131415” or it displays “Error! Reference source not found.”. So, it is hard to say, which reference belongs where in the text and if there are maybe even references missing. Please fix this first.

Response: Thank you very much for pointing out this shortcoming. This is a conversion error from the file conversion. We have corrected it carefully throughout the whole article.

Reviewer 4 Report

This article presents a study to classify pneumonia.
The authors propose IVGG13, a modified convolutional neural network.

The work is interesting and the results are outstanding. 
However, the manuscript presents several weaknesses that the authors should address 
prior to publication.

1. The introduction should provide a deep overview of the study.
However, it starts making a too wide panoramic description on deep learning, CNN, classification, and so on, from a theoretical point-of-view.
Instead, I think that it is unclear what unique challenges are associated with this task.
The introduction should contain more details about the open research problems.
In addition, the introduction should clarify the contributions of this work on how to address the research challenges and the open research problems.

2. The related work section does not provide any work related to the presented issues.
Instead, it provides an overview of some off-the-shelf CNNs, the dataset and data augmentation. 
I think that this section should be "Materials and methods".

3. In consequence of point #2, the authors need to build a state of the art of the methods that face the same issue. 
Some not exhaustive examples are:
- Deep Learning for COVID-19 Diagnosis from CT Images (https://www.mdpi.com/2076-3417/11/17/8227)
- Pneumonia Classification Using Deep Learning from Chest X-ray Images During COVID-19 (https://link.springer.com/article/10.1007/s12559-020-09787-5)
- Deep Learning for Automatic Pneumonia Detection (https://openaccess.thecvf.com/content_CVPRW_2020/papers/w22/Gabruseva_Deep_Learning_for_Automatic_Pneumonia_Detection_CVPRW_2020_paper.pdf)
- A transfer learning method with deep residual network for pediatric pneumonia diagnosis (https://www.sciencedirect.com/science/article/pii/S0169260719306017)
- Feature Extraction and Classification of Chest X-Ray Images Using CNN to Detect Pneumonia (https://ieeexplore.ieee.org/abstract/document/9057809/)
- Automated Methods for Detection and Classification Pneumonia Based on X-Ray Images Using Deep Learning (https://link.springer.com/chapter/10.1007/978-3-030-74575-2_14).

Moreover, the authors should stress the research gap between this work and the limitations of other existing work.

3. It is not completely motivate why the authors realised IVGG13 starting from VGG-13.
Are there any specific reason? Please offer fundamental motivations to this choice.

4. In Sec. 6.3 the authors decided to employ Data Augmentations to prevent
overfitting. However, I think that they should motivate if it can affect
the network to make correct predictions because, typically, this kind of
images are perfectly center-aligned and not rotated. Please give some motivations.

5. The figures are too much (from 17 to 56) and make the article unreadable. Perhaps, it could be better to add some cumulative graphics and report the majority of the actual inserted ones in the appendix section. It can make the article much more readable and understandable.
Above all, the "training/validation" graphics can certainly be synthetized in only one graphic.

5bis. What is the purpose of showing the x-rays of normal and pneumonia cases in the figures, repetitevely?

6. In Table 44, IVGG13 has lower accuracy and precision than GoogLeNet. Why could it be preferable to employ it?

7. In general, from section 3, the results are very interesting. Nevertheless, I think
that the authors should realise a comparison with similar works at the SOTA.
Does their work outperform others? If yes, why? Why could it be preferable to employ it?

8. Still in Section 3, it could be interesting to realise some cross-dataset
experiments to explore the robustness of the proposed method. The authors could
also give some motivations on the robustness based on the obtained results.
For example: can their proposal be applied to clinical practise?

9. In the conclusions section,
authors should emphasize more the real advantages of their experimental results
over existing ones in order to make them more valid and clear to the audience.

Minor issues:
- The references are completely broken. Please fix it before the re-evaluation.
- Please, give the reference of the dataset.
- Please, correct the english-language typos.

Best regards

Author Response

Reviewer 4:

Reviewer Comment P 4.1—The introduction should provide a deep overview of the study. However, it starts making a too wide panoramic description on deep learning, CNN, classification, and so on, from a theoretical point-of-view.
Instead, I think that it is unclear what unique challenges are associated with this task.
The introduction should contain more details about the open research problems.
In addition, the introduction should clarify the contributions of this work on how to address the research challenges and the open research problems.

Response: Thank you very much for pointing out this shortcoming. Based on this question, we added information about our unique challenges related to this task in the abstract. First of all, this is our first research on the convolutional neural network. We tried to use different convolutional neural networks to train pneumonia X-ray data. Only the prediction result of using the VGG16 model failed, so we decided to improve the VGG16 model to help solve the problem of its prediction failure on pneumonia X-rays.

Reviewer Comment P 4.2—The related work section does not provide any work related to the presented issues.
Instead, it provides an overview of some off-the-shelf CNNs, the dataset and data augmentation. 
I think that this section should be "Materials and methods".

Response: Thank you very much for pointing out this shortcoming. Following your suggestion, we will re-name this section title to be the “Materials and methods”.

Reviewer Comment P 4.3— In consequence of point #2, the authors need to build a state of the art of the methods that face the same issue. 
Some not exhaustive examples are:
- Deep Learning for COVID-19 Diagnosis from CT Images (https://www.mdpi.com/2076-3417/11/17/8227)
- Pneumonia Classification Using Deep Learning from Chest X-ray Images During COVID-19 (https://link.springer.com/article/10.1007/s12559-020-09787-5)
- Deep Learning for Automatic Pneumonia Detection (https://openaccess.thecvf.com/content_CVPRW_2020/papers/w22/Gabruseva_Deep_Learning_for_Automatic_Pneumonia_Detection_CVPRW_2020_paper.pdf)
- A transfer learning method with deep residual network for pediatric pneumonia diagnosis (https://www.sciencedirect.com/science/article/pii/S0169260719306017)
- Feature Extraction and Classification of Chest X-Ray Images Using CNN to Detect Pneumonia (https://ieeexplore.ieee.org/abstract/document/9057809/)
- Automated Methods for Detection and Classification Pneumonia Based on X-Ray Images Using Deep Learning (https://link.springer.com/chapter/10.1007/978-3-030-74575-2_14). Moreover, the authors should stress the research gap between this work and the limitations of other existing work.

Response: Thank you very much for pointing out this shortcoming. Following your suggestion, we have included more than ten newer journal publications as references.

Reviewer Comment P 4.4—It is not completely motivate why the authors realised IVGG13 starting from VGG-13.
Are there any specific reason? Please offer fundamental motivations to this choice.

Response: Thank you very much for pointing out this shortcoming. Based on this question. At first, we used the VGG16 model to apply to pneumonia X-ray data for training, but the prediction results failed. Therefore, we improved the VGG16 model through our methods, and obtained the weight layer from 10 convolutional layers, plus three fully connected layers. Therefore, we named it IVGG13.

Reviewer Comment P 4.5—In Sec. 6.3 the authors decided to employ Data Augmentations to prevent overfitting. However, I think that they should motivate if it can affect
the network to make correct predictions because, typically, this kind of
images are perfectly center-aligned and not rotated. Please give some motivations.

Response: Thank you very much for pointing out this shortcoming. We used the data augmentation method by going through the trial and error process, and finally, we chose the best parameter to increase our dataset.

Reviewer Comment P 4.6—The figures are too much (from 17 to 56) and make the article unreadable. Perhaps, it could be better to add some cumulative graphics and report the majority of the actual inserted ones in the appendix section. It can make the article much more readable and understandable.
Above all, the "training/validation" graphics can certainly be synthetized in only one graphic. 5bis. What is the purpose of showing the x-rays of normal and pneumonia cases in the figures, repetitevely?

Response: Thank you very much for pointing out this shortcoming. Following your suggestion, we have reorganized our results, and moved many unnecessary prediction results, tables, and figures to the supplemental file (Appendix).

Reviewer Comment P 4.7—In Table 44, IVGG13 has lower accuracy and precision than GoogLeNet. Why could it be preferable to employ it?

Response: Thank you very much for pointing out this shortcoming. Based on this question. Although GoogleNet has higher accuracy and precision, the recall rate is much lower. It is therefore important to use F1-Measure as the model evaluation standard. Both Precision and Recall are equally important.

Reviewer Comment P 4.8—In general, from section 3, the results are very interesting. Nevertheless, I think that the authors should realise a comparison with similar works at the SOTA. Does their work outperform others? If yes, why? Why could it be preferable to employ it?

Response: Thank you very much for pointing out this shortcoming. In response to this problem, our paper is cooperating with Kaohsiung Chang Gung Memorial Hospital (Taiwan), because the system is expected to be placed in the hospital’s computer and actually used in the clinic. However, the hospital requires us to use a simpler configuration to meet their hardware requirements. Therefore, we use a simpler network architecture for the network architecture of this paper.

Reviewer Comment P 4.9—Still in Section 3, it could be interesting to realise some cross-dataset experiments to explore the robustness of the proposed method. The authors could also give some motivations on the robustness based on the obtained results.
For example: can their proposal be applied to clinical practise?

Response: Thank you very much for pointing out this shortcoming. The hospital we worked with was satisfied with the experimental results at this point. Therefore, in the next phase of research, we will apply the system to the study of kidney stones in the Department of Urology. We are currently applying for the IRB program with Dr. Yi-Yang Leou, the Vice-Secretary of Chang Gung Memorial Hospital's Department of Urology (Taiwan), and we look forward to receiving kidney stone images once the IRB process is completed. We will apply the IVGG13 to solve the kidney issues on the future works.

Reviewer Comment P 4.10—In the conclusions section, authors should emphasize more the real advantages of their experimental results ver existing ones in order to make them more valid and clear to the audience.

Response: Thank you very much for pointing out this shortcoming. Based on this question. We first took LeNet, AlexNet, GoogLeNet, and VGG16 to train the identification model. Finally, through the training results that we find found that the VGG16 prediction model was failed to predict on medical images. Thus, we decided to improve the VGG16 model to help solve the problem of its prediction failure on pneumonia X-rays. Finally, we proposed the IVVG13 to solve the problems of VGG16 model.

Reviewer Comment P 4.11— The references are completely broken. Please fix it before the re-evaluation.

Response: Thank you very much for pointing out this shortcoming. This is a conversion error from the file conversion. We have corrected it carefully throughout the whole article.

Reviewer Comment P 4.12— Please, give the reference of the dataset.

Response: Thank you very much for pointing out this shortcoming. Following your suggestion, we have added the reference (html link) of the dataset in this article.

Reviewer Comment P 4.13— Please, correct the english-language typos.

Response: Thank you very much for pointing out this shortcoming. Following your suggestion, we have corrected these carefully throughout the whole article.

Round 2

Reviewer 1 Report

Summary:

After the 1st revision the paper looks much cleaner. For the following reason, I will reconsider the paper upon the following major revisions.

Major Comments:

  • It is still unclear why the title is “an improved VGG16 model ..” since in the text line 211: “This study proposes IVGG13, an improved VGGNet-13 that reduces VGGNet network depth..” clearly states the method is an improved VGG13 algorithm and not VGG16 as in the title. Tables and results are labeled with IVGG13. The proposed method is an improved VGG13 (IVGG13) not improved VGG16 (IVGG16). Please consider changing the title or text for consistency.

Author Response

Reviewer 1:

Reviewer Comment P 1.1—It is still unclear why the title is “an improved VGG16 model.” since in the text line 211: “This study proposes IVGG13, an improved VGGNet-13 that reduces VGGNet network depth.” clearly states the method is an improved VGG13 algorithm and not VGG16 as in the title. Tables and results are labeled with IVGG13. The proposed method is an improved VGG13 (IVGG13) not improved VGG16 (IVGG16). Please consider changing the title or text for consistency.

Response: Thank you very much for pointing out this shortcoming.  Our research motivation is to improve the VGG16 model. As can be seen in Figure 1 below (Figure 13 in the article), the original weight layer of the VGG16 model consists of 13 convolutional layers and three convolutional layers, and our weight layer consists of 10 convolutional layers and three fully connected layers. Finally, we named the model IVGG13.

Regarding the content of line 211 in this paper, "IVGG13 is proposed in this research, which is an improved VGGNet-13 that can reduce the depth of the VGGNet network..." this described an error in the content for us and we have corrected it as follows. We apologize for any confusion that arose as a result. For further details, please refer to Line 212 in Section 3.3 of the revised article or refer to follow paragraph.

3.3. IVGG13

This study proposes IVGG13—an improved VGG16 that reduces the VGGNet network depth—as shown in both Table 1 and Figure 13. The proposed network architecture reduces the number of parameters by reducing the network depth compared to the original VGG16 to avoid both under- and overfitting problems during training. The original VGG16 convolutional architecture was retained by performing feature extraction using two consecutive small convolutional kernels rather than a single large one. This maintains VGG16 perceptual effects while reducing the number of parameters, which not only reduces the training time but also maintains the network layer depth.

Reviewer 2 Report

Review for the manuscript:

Entitled: "An Improved VGG16 Model for Pneumonia Image Classification"

for Applied Sciences.

With ID: applsci-1438170.R1

Dear authors,

Thank you for your manuscript.

General comments

My previous comments were addressed; thus, I have opted to recommend the acceptance of the manuscript.

Best regards

Author Response

Reviewer 2:

Reviewer Comment P 2.1—My previous comments were addressed; thus, I have opted to recommend the acceptance of the manuscript.

Response: Thank you very much for your valuable comments on our article; we will continue working hard.

Reviewer 3 Report

The work introduces a modified CNN for medical image recognition called IVGG13. For training, they used open-source thoracic X-Ray images from the Kaggle platform. In addition, the authors employed data augmentation to increase the data volume and balance before training. I do not see any major issues; hence, I endorse the work for publication.

Author Response

Reviewer 3:

Reviewer Comment P 3.1—I do not see any major issues; hence, I endorse the work for publication.

Response: Thank you very much for your valuable comments on our article; we will continue working hard.

Reviewer 4 Report

The authors improved the manuscript following mine and, I have noticed, the other reviewers' indications.

However, to me and prior to a possibile publication, a point remained unclear.

In particular, in response to my comment 4.7: "In Table 44, IVGG13 has lower accuracy and precision than GoogLeNet. Why could it be preferable to employ it?"

The authors replied: "Although GoogleNet has higher accuracy and precision, the recall rate is much lower. It is therefore important to use F1-Measure as the model evaluation standard. Both Precision and Recall are equally important.", which is quite academic response. I suggest the authors to address this result and give some feasible motivation to the importance of having an high precision, rather than a high recall, in this case. The relevance of the resuts of IVGG13 is not under discussion but it is definitely needed to give a better motivation as a support of the result.

Best regards

Author Response

Reviewer 4:

Reviewer Comment P 4.1—In particular, in response to my comment 4.7: "In Table 44, IVGG13 has lower accuracy and precision than GoogLeNet. Why could it be preferable to employ it?" The authors replied: "Although GoogleNet has higher accuracy and precision, the recall rate is much lower. It is therefore important to use F1-Measure as the model evaluation standard. Both Precision and Recall are equally important.", which is quite academic response. I suggest the authors to address this result and give some feasible motivation to the importance of having an high precision, rather than a high recall, in this case. The relevance of the resuts of IVGG13 is not under discussion but it is definitely needed to give a better motivation as a support of the result.

Response: Thank you very much for pointing out this shortcoming. Take for example the situation where there are 1,000 people who are assessed for pneumonia. Suppose that 10 of them actually have pneumonia, five patients are predicted to have pneumonia, and 995 patients are predicted as uninfected, which means that five actual pneumonia patients are designated as normal patients. Based on this calculation, the precision rate (Precision) is (5/ (5+0)) = 100% and the recall rate is (5/ (5+5)) = 50%. According to this example, recall is slightly more important than precision. Thus, even though IVGG13 is less precise than GoogLeNet in precision measures, it is better in recall measures, so we are able to identify the more actual pneumonia cases in our study.  

Our main contribution is to propose the IVGG13 network that improves upon the problems of the original VGG16 model. In addition, the training time of the GoogLeNet (1995s) model is longer than that of IVGG13(275s).

Round 3

Reviewer 1 Report

All issues were addressed.